# Acute Hypoxia Profile is a Stronger Prognostic Factor than Chronic Hypoxia in Advanced Stage Head and Neck Cancer Patients

**DOI:** 10.3390/cancers11040583

**Published:** 2019-04-25

**Authors:** Martijn van der Heijden, Monique C. de Jong, Caroline V. M. Verhagen, Reinout H. de Roest, Sebastian Sanduleanu, Frank Hoebers, C. René Leemans, Ruud H. Brakenhoff, Conchita Vens, Marcel Verheij, Michiel W. M. van den Brekel

**Affiliations:** 1Division of Cell Biology, The Netherlands Cancer Institute, Plesmanlaan 121, 1006 CX Amsterdam, The Netherlands; ma.vd.heijden@nki.nl (M.v.d.H.); cverhagen@rijnstate.nl (C.V.M.V.); c.vens@nki.nl (C.V.); m.verheij@nki.nl (M.V.); 2Department of Head and Neck Oncology and Surgery, The Netherlands Cancer Institute, Plesmanlaan 121, 1006 CX Amsterdam, The Netherlands; 3Department of Radiation Oncology, The Netherlands Cancer Institute, Plesmanlaan 121, 1006 CX Amsterdam, The Netherlands; m.d.jong@nki.nl; 4Amsterdam UMC, Vrije Universiteit Amsterdam, Otolaryngology/Head & Neck Surgery, Cancer Center Amsterdam, De Boelelaan 1117, 1081 HV Amsterdam, The Netherlands; r.deroest@vumc.nl (R.H.d.R.); cr.leemans@vumc.nl (C.R.L.); rh.brakenhoff@vumc.nl (R.H.B.); 5Department of Radiation Oncology (MAASTRO), GROW – School for Oncology and Developmental Biology, Maastricht University Medical Centre, Dr. Tanslaan 12, 6229 ET Maastricht, The Netherlands; s.sanduleanu@maastrichtuniversity.nl (S.S.); frank.hoebers@maastro.nl (F.H.); 6Department of Radiation Oncology, Radboud University Medical Center, Geert Grooteplein Zuid 10, 6525 GA Nijmegen, The Netherlands; 7Department of Oral and Maxillofacial Surgery, Amsterdam UMC, Academic Medical Center, Meibergdreef 9, 1105 AZ Amsterdam, The Netherlands

**Keywords:** head and neck cancer, gene expression, hypoxia

## Abstract

Hypoxic head and neck tumors respond poorly to radiotherapy and can be identified using gene expression profiles. However, it is unknown whether treatment outcome is driven by acute or chronic hypoxia. Gene expression data of 398 head and neck cancers was collected. Four clinical hypoxia profiles were compared to in vitro acute and chronic hypoxia profiles. Chronic and acute hypoxia profiles were tested for their association to outcome using Cox proportional hazard analyses. In an initial set of 224 patients, scores of the four clinical hypoxia profiles correlated with each other and with chronic hypoxia. However, the acute hypoxia profile showed a stronger association with local recurrence after chemoradiotherapy (*p* = 0.02; HR = 3.1) than the four clinical (chronic hypoxia) profiles (*p* = 0.2; HR = 0.9). An independent set of 174 patients confirmed that acute hypoxia is a stronger prognostic factor than chronic hypoxia for overall survival, progression-free survival, local and locoregional control. Multivariable analyses accounting for known prognostic factors substantiate this finding (*p* = 0.045; *p* = 0.042; *p* = 0.018 and *p* = 0.003, respectively). In conclusion, the four clinical hypoxia profiles are related to chronic hypoxia and not acute hypoxia. The acute hypoxia profile shows a stronger association with patient outcome and should be incorporated into existing prediction models.

## 1. Introduction

The average overall survival for advanced stage head and neck cancer patients is around 50% [1], but this varies greatly among different groups of patients. Clinical (TNM) staging explains survival variation only partially for these patients [2,3,4]. Human papillomavirus (HPV) positive oropharyngeal tumors represent a distinct subgroup of head and neck squamous cell carcinoma (HNSCC) that is associated with a good prognosis [5]. We have previously shown that the addition of HPV status and a prognostic gene expression profile can improve outcome prediction, suggesting that a substantial part of the survival variation is explained by tumor biology [5].

Hypoxia is one of the most studied biological factors affecting prognosis in HNSCC [6]. Tumor cells can become hypoxic by chronic (diffusion limited) and acute (perfusion limited) mechanisms [7]. Perfusion or diffusion limited, intermittent or cycling hypoxia are alternative terms that have been used to better reflect the mechanisms that result in acute hypoxia as referred to in this study. Both acute and chronic hypoxia can have different effects on tumor cells and their microenvironment. It is unclear whether prognosis is mostly impacted by chronic or acute hypoxia [7]. Because oxygen is essential for DNA-damage upon irradiation, hypoxic tumors respond poorly to radiation [8,9]. Since two third of all HNSCC patients receive radiotherapy as part of their treatment, hypoxia can reduce efficacy of the treatment and impact prognosis [10]. A meta-analysis of clinical trials showed that in vivo modification of the hypoxia status during radiotherapy improves survival of HNSCC patients, demonstrating that hypoxia is an important factor in radioresistance [11]. Likewise, hypoxia-inducible factor is a poor prognostic factor in surgically treated HNSCC patients [12]. Unfortunately, the benefit from hypoxia modification therapy was modest and comes with added toxicity [11]. This led to the hypothesis that only patients with hypoxic tumors profit from such a therapeutic intervention, which was further confirmed in two subsequent studies [13,14]. Initiated in 2014, the NIMRAD study aims to ‘prospectively validate a hypoxia gene signature that can be used in clinical practice to personalize treatment and select appropriate patients for hypoxia modifying treatment’ [15]. 

A robust method to quantify hypoxia in tumors is required to select patients for hypoxia modifying treatment. Different techniques have been applied to evaluate the level of hypoxia in a tumor and its impact on radiotherapy response [6], including an oxygen-sensitive needle probe inserted into the tumor [16,17,18,19], exogenous immunohistochemical markers (e.g., pimonidazole [20]), endogenous biomarkers (e.g., HIF1-alpha [21,22] or carbonic anhydrase IX [14,21]) and imaging techniques like MRI [23] and PET [24]. None of these techniques are currently used in routine clinical practice because they are too complex or insufficiently accurate. Tumor hypoxia can also be deduced from gene expression profiling and many gene expression profiles for hypoxia have been proposed [13,21,25,26,27,28,29]. 

In HNSCC, four hypoxia profiles have been validated to be prognostic or predictive [13,25,26,27]. Winter et al. [25] obtained a hypoxia 99-gene expression profile that was validated to be associated with recurrence free survival. Buffa et al. [26] used hypoxia-regulated genes to select co-expressed genes in three HNSCC and five breast cancer studies. The resulting 51-gene profile was validated for its prognostic relevance in four independent datasets. Toustrup et al. [13] generated a profile from in vitro experiments together with gene expression data from 58 head and neck cancer biopsies. The resulting 15-gene expression profile proved to be predictive for the response to nimorazole during radiotherapy in 323 HNSCC patients. Eustace et al. [27] reduced the profile from Buffa et al. to a 26-gene profile that predicted regional control in 157 laryngeal cancer patients treated with radiotherapy. Some authors suggest that the use of multiple hypoxia profiles improves the prognostic value [30]. However, none of these studies made a distinction between acute and chronic hypoxia. Gene expression profiles for acute and chronic hypoxia were generated in vitro. Seigneuric et al. used temporal changes in human epithelial mammary cell lines in response to hypoxia to generate gene expression profiles for acute and chronic hypoxia and showed that acute hypoxia is a prognostic factor in breast cancer [31].

In this study we aimed to improve hypoxia-based HNSCC patient prognostication by evaluating the contribution of chronic and acute hypoxia in gene expression profiles. We selected the four clinical HNSCC hypoxia gene expression profiles which are associated with patient outcome in HNSCC [13,25,26,27]. In an initial set of 224 patients, these gene expression profiles were compared amongst each other and to the in vitro chronic and acute hypoxia gene expression profiles of Seigneuric et al. [31]. The prognostic value of these acute and chronic hypoxia expression profiles was then tested in a new cohort of 91 HNSCC patients and validated in an independent cohort of 174 HPV-negative HNSCC patients, all treated with definitive cisplatin based chemoradiotherapy. Lastly, the prognostic value of acute and chronic hypoxia expression profiles was tested in a multivariable analysis with known prognostic factors in HNSCC.

## 2. Results

### 2.1. Few Overlapping Genes in the Four Clinical HNSCC Hypoxia Gene Expression Profiles

We compared the four clinical gene expression profiles of Winter et al., Buffa et al., Toustrup et al. and Eustace et al. to acute and chronic hypoxia profiles of Seigneuric et al. (Figure 1). A more extensive overview of the profiles is presented in Appendix A. The four clinical profiles consisted of 147 unique genes. Of these, 82% was present in only one of the four profiles. Three genes (2%) were present in all four signatures: Aldolase A(ALDOA), Prolyl 4-Hydroxylase Subunit Alpha 1 (P4HA1) and Solute Carrier Family 2 Member 1 (SLC2A1 a.k.a. GLUT-1). Aldolase A is a glycolytic enzyme, the P4HA1 gene encodes a component of a key enzyme in collagen synthesis and the SLC2A1 (a.k.a. GLUT-1) gene encodes a glucose transporter. None of these three genes were present in the Seigneuric chronic in vitro profile, however 9 genes in this profile were present in at least one of the four clinical profiles. The Seigneuric acute profile had no overlapping genes with any of the clinical profiles or the Seigneuric chronic in vitro profile. 

### 2.2. The Four Clinical HNSCC Hypoxia Gene Expression Profiles are Correlated and Resemble Chronic Hypoxia Response

We tested the conformity of the four clinical hypoxia profiles in 224 HNSCC patients. To this end, we combined gene expression data generated from three HNSCC cohorts (Table 1): Pramana (published), de Jong 1 (expanded expression data set with additional samples that could not be used in the original study) and unpublished gene expression data from a patient cohort as described in de Jong 2 et al. [32,33,34,35,36]. Patient characteristics of the cohorts are available in Appendix A. Scores for the four clinical profiles and the Seigneuric acute and chronic hypoxia profiles were generated for all 224 HNSCC patients. The average Spearman correlation between the scores of the four hypoxia profiles was 0.82 and highly significant (range 0.71–0.90, *p*-values < 0.0001, Figure 2A), demonstrating that the four gene expression profiles rank patients similarly. All four clinical hypoxia profiles were significantly correlated to the Seigneuric chronic hypoxia profile (correlation 0.60, *p* < 0.0001, Figure 2A,). The average correlation with Seigneuric acute hypoxia was −0.09 (*p* = 0.2). A clustering based on all scores showed that the four clinical HNSCC hypoxia profiles cluster together with the Seigneuric chronic profile, whereas no correlation was observed with the Seigneuric acute profile (Figure 2B). Together this strongly suggests that the four clinical profiles represent chronic hypoxia and not acute hypoxia. 

### 2.3. The Acute Hypoxia Profile is Associated with Local Control

As the inclusion criteria of the three cohorts were not comparable, we did not attempt to combine the three cohorts for outcome analyses. The largest cohort with Local Control (LC) data available (Pramana cohort, *n* = 91) was used to assess the prognostic value of acute and chronic hypoxia [32]. Unfortunately, no clinically validated acute hypoxia profiles are available for HNSCC. We therefore used the Seigneuric acute hypoxia profile as a surrogate marker for acute hypoxia in HNSCC. Since the four clinically validated HNSCC hypoxia profiles were highly correlated to each other and to the Seigneuric Chronic profile (Figure 2A), for each patient the scores of the four clinical profiles were averaged to obtain a joint chronic hypoxia score. The median was used to define “High” and “Low” hypoxia groups. 

The acute hypoxia profile was significantly associated with local control (*p* = 0.02). Patients in the acute hypoxia “High” group showed a higher local recurrence rate (Hazard Ratio (HR): 3.1, 95%CI: 1.1−8.6), compared to patients in the acute hypoxia “Low” group. Patients in chronic hypoxia “High” group tended to have a worse local control rate (*p* = 0.2, HR = 1.9; Figure 3). Kaplan Meier curves and hazard ratios for the individual profiles are provided in Appendix A. To study the combined effect of the acute hypoxia profile and the chronic hypoxia profile we generated three groups: (1) acute and chronic hypoxia profile both low, (2) acute or chronic hypoxia profile high and (3) acute and chronic hypoxia profile both high (Figure 3C). The comparison in Figure 3C shows that, local control rates are worst when both acute and chronic hypoxia profiles are high (*p* = 0.04).

### 2.4. Validation of the Acute Hypoxia Profile as a Prognostic Marker for Outcome in HNSCC

HPV is a major prognostic factor in HNSCC and the Pramana cohort contained HPV-positive and HPV-negative tumors. So, to test whether the prognostic value of acute hypoxia is independent of HPV, we aimed to validate the prognostic relevance of acute hypoxia in a HPV-negative HNSCC cohort. To this end, we collected clinical and gene expression data from a study by Van der Heijden et al. [35,36]. This data derived from pre-treatment patient tumor material of two cohorts of HPV-negative advanced stage HNSCC patients, NKI-CRAD and DESIGN, comprising 174 patients in total. All patients were treated with chemo-radiotherapy and an overview of patient characteristics is available in Appendix A. Overall Survival (OS), Progression Free Survival (PFS), Local Control (LC), Locoregional Control (LRC) and Distant Metastasis free survival (DM) data were available for this dataset. As before, scores for all 174 patients were calculated. The Seigneuric acute hypoxia profile was used as a marker for acute hypoxia and the scores of the four clinical HNSCC hypoxia profiles (Winter et al., Buffa et al., Toustrup et al. and Eustace et al.) were averaged to obtain a joint chronic hypoxia score. The distribution of the profiles is shown in Appendix A. Median splits were used to divide patients in “High” and “Low” groups. 

Confirming the findings in the Pramana cohort, the acute hypoxia “High” group had a significantly worse LC (*p* = 0.006, HR = 3.3; Figure 4A), OS (HR = 1.58, *p* = 0.023), PFS (HR = 1.66, *p* = 0.009) and LRC; (HR = 2.4, *p* = 0.008; Appendix A). The chronic hypoxia “High” group showed a significant difference in PFS only (HR = 1.52, *p* = 0.03) and a trend for OS (HR = 1.43, *p* = 0.075; supplementary Materials, Appendix A), but no significant difference in LC (*p* = 0.34; HR = 1.43; Figure 4B) and LRC (HR = 1.59, *p* = 0.141). We next combined the chronic hypoxia and acute hypoxia groups as described earlier to obtain three groups. When both acute and chronic hypoxia expression profiles are “High”, OS, PFS, LC and LRC rates are significantly worse than when both are low (Figure 4C–E and Appendix A). 

### 2.5. Multivariable Analysis Confirms the Prognostic Value of Acute Hypoxia

To determine whether acute hypoxia has additional prognostic value to known prognostic markers, other than HPV, we performed a multivariable in the Van der Heijden cohort. For 149 patients, tumor volume data were available. Within these 149 patients all available variables were tested in a univariable analysis for their association with OS, PFS, LC, LRC and DM (Appendix A). To avoid arbitrary cut-offs, acute hypoxia, chronic hypoxia (the average of the four clinical hypoxia scores) and tumor volume were tested as continuous variables. 

Acute hypoxia, gender, tumor site, disease stage, cisplatin dose, and tumor volume showed significant associations with multiple patient outcome measures in univariable analysis (Appendix A). Chronic hypoxia only showed associations with locoregional control. Variables with significant associations to patient outcome were combined in a multivariable Cox proportional hazard model (Table 2). Previous research has shown an interaction between hypoxia and tumor volume [37], this interaction was therefore incorporated in the multivariable analysis. Results of the multivariable analysis show that acute hypoxia is significantly associated with OS (HR = 3.5, *p* = 0.045), PFS (HR = 3.3, *p* = 0.042), LC (HR = 38.2, *p* = 0.018) and LRC (HR = 27, *p* = 0.003), whereas chronic hypoxia is not associated with any of the outcomes. Tumor volume (per cm^3^ increase) is also associated with OS (HR = 1.01, *p* = 4.12E-05), PFS (HR = 1.008, p = 0.0007) and LC (HR = 1.014, *p* = 0.027). The interaction of acute hypoxia with tumor volume was significantly associated with OS (HR = 0.98, *p* = 0.004), PFS (HR = 0.98, *p* = 0.02), and LRC (HR = 0.95, *p* = 0.016). The interaction of chronic hypoxia with tumor volume was not significantly associated with any of the reported outcomes. These analyses confirm the prognostic value of acute hypoxia independent from clinical factors, tumor volume and chronic hypoxia.

## 3. Discussion

The four clinical gene expression profiles for hypoxia, which have been validated to predict outcome in HNSCC, have few overlapping genes. Nevertheless, they classified patients similarly, indicating that they reflect a similar underlying biological process: chronic and not acute hypoxia. Since the clinical hypoxia profiles were correlated, we combined them in a joint chronic hypoxia score. In contrast to the chronic hypoxia profile, the acute profile was associated with local control in the Pramana cohort. The poor association of the acute hypoxia profile with patient outcome was validated in an independent validation cohort (*n* = 174) and multivariable analysis showed that acute hypoxia is a significant prognostic factor independent of clinical factors, tumor volume and chronic hypoxia. 

The phenomenon that different gene expression profiles, with different genes, can describe the same process, has been reported before [38]. Given the fact that over 4,000 genes are hypoxia-influenced, it seems reasonable to assume that multiple robust, but different, hypoxia gene expression profiles can be assembled [39]. It should be noted that all gene expression profiles have been applied with the same method in our study. Due to lack of access to the reference cohort, it was not always possible to apply gene expression profiles as in the original publication. To be able to compare the gene expression profiles, we decided to apply the same method to all gene expression profiles.

### 3.1. Acute and Chronic Hypoxia

The terms acute and chronic hypoxia are simplified terms to describe a complex spectrum of hypoxic micro environmental alterations in a tumor [40]. While an absolute distinction between the two cannot be made, many suggestions for the separate origin, measurement and treatment of the two entities have been published [41,42,43,44]. Janssen et al. employed various staining protocols to study acute and chronic hypoxia in head and neck tumors [42]. They showed that tumors contained on average 15% acute hypoxic (proliferating cells around non-perfused vessels) and around 30% chronic hypoxic areas (cells at a large distance from blood vessels). The two different areas did not overlap. Also in vitro studies showed that cells that had been under hypoxia for a short time, showed a different gene expression profile compared to cells that were hypoxic for a prolonged time [31]. As described by Lendahl et al. 4,047 genes were hypoxia-regulated in a colon carcinoma cell line. Only 52 genes were specific to the acute (1 or 2 h) hypoxia response. 144 genes were up- or downregulated by both acute and chronic (24 h) hypoxia, whereas the majority of the genes (4,005) were chronic hypoxia specific [39]. Nonetheless, in the past decades research has focused on generating a gene expression profile for ‘hypoxia’ in general, without distinction of acute and chronic. The fact that all four clinical HNSCC hypoxia profiles correlated with chronic hypoxia could be due to the large excess of genes regulated by chronic hypoxia [39]. Also, the methods used to generate the gene expression profile could be an explanation that the profiles correlated with chronic hypoxia. For example, the Toustrup et al. profile is based on Eppendorf probe measurements to find relevant genes. If indeed, the area of chronic hypoxia is on average twice the area of acute hypoxia, as reported by Janssen et al., this could lead to a bias towards genes that are correlated with chronic hypoxia. Winter, Buffa and Eustace et al. started with 10 hypoxia ‘seed genes’ to develop their signatures. In our data, these 10 genes were not correlated with in vitro Seigneuric acute hypoxia but showed correlations to the in vitro Seigneuric chronic hypoxia profile (Appendix A).

### 3.2. Acute Hypoxia and Prognosis

The importance of acute hypoxia has been recognized for decades [45]. For example, Chan et al. showed that a human lung squamous cell carcinoma cell line (H1299) becomes more radioresistant under acute hypoxia than under chronic hypoxia, with respective oxygen enhancement ratios of 1.96 and 1.37 [46]. Unfortunately, conclusive data on the separate and combined prognostic effects of acute and chronic hypoxia in HNSCC are lacking. This might be due to the fact that it is difficult to distinguish both types of hypoxia with immunohistochemistry. We found that patients with high acute or chronic hypoxia expression, had a 3.1 and 1.9 times higher risk of a local recurrence in the Pramana cohort, respectively. This was confirmed in the validation set of 174 patients with hazard ratios of 3.3 and 1.4 for local failure, for acute and chronic hypoxia, respectively. Although in both sets chronic hypoxia was not significant, the effect size appears comparable to previously reported hazard ratios for chronic hypoxia [13,25,26,27]. For chronic hypoxia gene expression profiles, the general deduction is that more hypoxic tumors are approximately twice as likely to recur than less hypoxic tumors. This effect could be underestimated due to a division into two hypoxia groups according to the median. We therefore used the scores as continuous variables in the multivariable analysis, which shows that acute hypoxia is a stronger prognostic factor than chronic hypoxia. Literature has suggested that cells lacking functional p53 are more susceptible to genomic instability and potential tumorigenesis when experiencing reoxygenation after acute hypoxia compared to chronic hypoxia [47]. This might also explain why the effect is more pronounced in the Van der Heijden cohort, which contains only HPV-negative HNSCC in which p53 mutations are highly prevalent. Since acute and chronic hypoxia have a different etiology, knowledge about which type of hypoxia causes radioresistance in a specific patient, could lead to the use and development of strategies targeting acute or chronic hypoxia.

## 4. Materials and Methods

### 4.1. Gene Expression Profiles

We selected four gene expression profiles for hypoxia that are clinically validated for head and neck cancer, namely, Winter et al., Buffa et al., Toustrup et al. and Eustace et al. Gene expression profiles generated by Seigneuric et al. were used for in vitro acute and chronic hypoxia. Since 2% oxygen is relatively close to oxygen concentration in normal tissue we used the <0.02% oxygen derived profiles as the in vitro acute and chronic hypoxia profiles. There is no clear cut-off for acute and chronic hypoxia. To simplify the complex spectrum of hypoxia, we used the Seigneuric 1 h time point profile for acute hypoxia, while the Seigneuric 24 h time point profile was used for chronic hypoxia [6].

### 4.2. Datasets

We initially collected gene expression data of three different HNSCC patient cohorts (Pramana, De Jong 1 and De Jong 2), comprising 224 patients in total (Table 1). Extensive patient characteristics for each cohort are available in the Appendix A and the original publications [32,33,34]. Clinical and expression data of Pramana et al. has been previously published. Briefly, the dataset of Pramana et al. consisted of 91 HPV-positive and negative HNSCC patients all treated with definitive cisplatin based chemoradiotherapy [32]. Gene expression data were obtained from fresh-frozen pre-treatment material, analyzed using dual-channel Operon microarray slides. The full dataset is available at (GEO database) and extensive methods are available in the original publication. 

Data of 52 matched patients of the De Jong 1 cohort was previously published [33]. This dataset was expanded with an additional 47 samples that could not be used in the original study. Briefly, gene expression of 99 fresh-frozen laryngeal and oropharyngeal (HPV status unknown) carcinomas was determined using the Illumina beads microarray platform. All patients were treated with radiotherapy. The full dataset is available at (GEO database, We are in the process of submitting to GEO and will provide the numbers a.s.a.p.) and has been generated as described previously.

Clinical data from the De Jong 2 cohort are as described previously [34]. The cohort consists of 34 laryngeal carcinomas treated with radiotherapy. Collection of tumor material and clinical data were approved by the Institutional Review Board of the Netherlands Cancer Institute and all patients signed informed consent. RNA was extracted from pre-treatment FFPE biopsies using the Roche High Pure miRNA Isolation Kit (REF: 05080576001). Quality and quantity of the total RNA was assessed with the 2100 Bioanalyzer using a Nano chip. TruSeq cDNA libraries were generated using the TruSeq RNA Library Preparation Kit v2 sample preparation kit (Illumina Cat.No RS-122-2001/2). The reads (51bp) were sequenced on a HiSeq2000 using V3 chemistry (Illumina Inc., San Diego CA, USA), aligned against the human genome, build 37, using Tophat (version 2.0.6). HTseq count was used to count the number of reads per gene. Only uniquely mapped reads were counted. Detailed methods are provided in the Appendix A. Data are available at http://ega-archive.org.

To validate our findings, we used gene expression data collected and generated within a separate study as described by van der Heijden et al. and deposited at http://ega-archive.org [35,36]. The “van der Heijden cohort” combines two independent cohorts, the NKI-CRAD and the DESIGN cohort, and comprises 174 HPV negative advanced stage HNSCC patients. All patients were treated with definitive cisplatin-based chemoradiotherapy. RNA was isolated from fresh-frozen material and gene-expression was determined using RNA-sequencing as described previously.

### 4.3. Data Analysis

All analyses were performed in R 3.4.3 using RStudio 1.1. The gene expression profiles consisted of genes exclusively upregulated under hypoxia. All expression data were normalized as described in the original publications. In the Pramana and both De Jong cohorts, per sample hypoxia scores were calculated for each gene expression profile using the mean normalized expression of all genes in the profile. In order to combine the three different patient cohorts, scores were rank-normalized per profile between 0 and 1 for each patient cohort. Gene-expression scores were calculated using Gene Set Variation Analysis (GSVA) [48] in the van der Heijden dataset. The median was used to split patients into “High” and “Low” hypoxia groups. Cox proportional hazard analyses were used to test associations with patient outcome (Overall Survival (OS), Progression Free Survival (PFS), Local Control (LC), Locoregional Control (LRC) and Distant Metastasis (DM)). Multivariable Cox proportional hazard analyses were performed in the Van der Heijden cohort and no cut-offs were used to test the continuous variables. 

## 5. Conclusions

All four clinical HNSCC hypoxia profiles correlate with chronic and not with acute hypoxia expression profiles. However, the acute hypoxia profile has a stronger association with patient outcome after chemoradiotherapy in both the test and validation cohort. Multivariable analysis shows the value of acute hypoxia in addition to well-known prognostic factors for HNSCC. Acute hypoxia gene expression should therefore be incorporated into existing hypoxia-based prediction models.

## Figures and Tables

**Figure 1 cancers-11-00583-f001:**
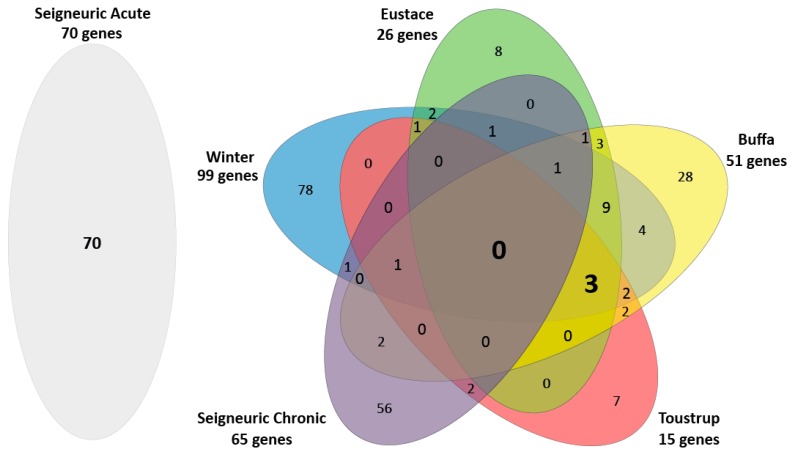
Venn diagram showing the number of overlapping genes in the four clinical hypoxia profiles and the Seigneuric acute and chronic in vitro profiles.

**Figure 2 cancers-11-00583-f002:**
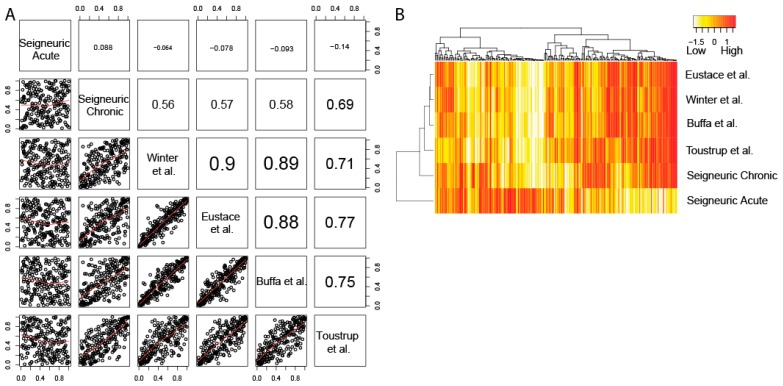
Correlations and clustering of the hypoxia scores from the clinical hypoxia profiles and the Seigneuric acute and chronic in vitro hypoxia profiles in 224 HNSCC patients. (**A**) Spearman correlations (upper right panels) and scatter plots (lower left panels) of all possible pairs of hypoxia profiles for 224 patients. All Spearman correlations were significant at the *p* < 0.0001 level. (**B**) Heatmap showing the scores for the four hypoxia profiles and in vitro profiles as indicated in 224 patients.

**Figure 3 cancers-11-00583-f003:**
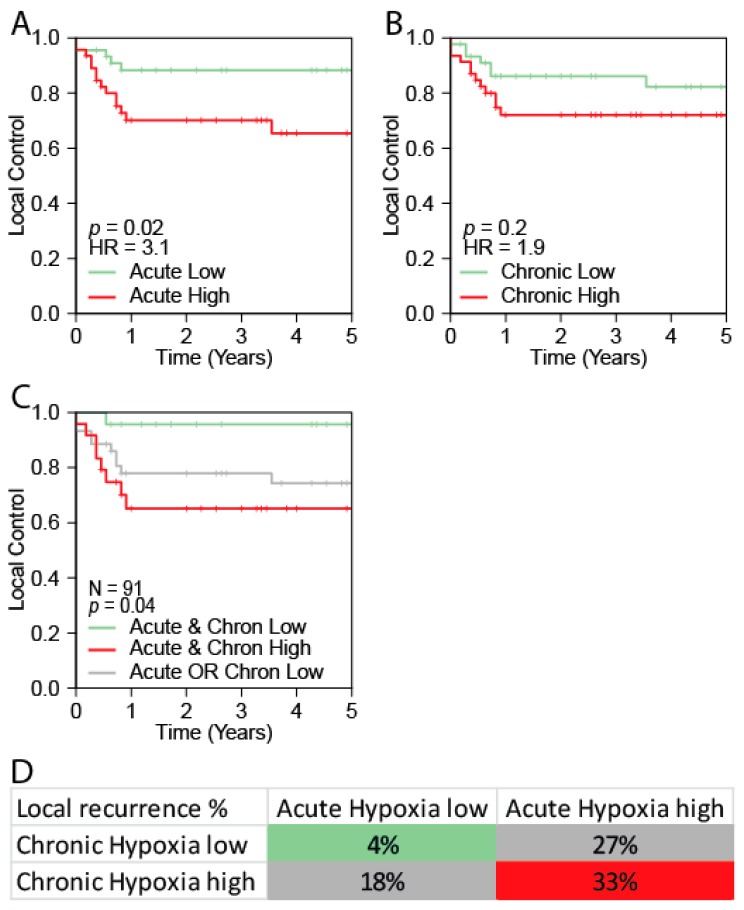
Combined acute and chronic hypoxia scores. (**A**) Kaplan–Meier curve showing local control of the “High” versus “Low” groups of the acute hypoxia profile. (**B**) Kaplan–Meier curve showing local control of the “High” versus “Low” groups of the chronic hypoxia profile. (**C**) Kaplan–Meier curve showing local control for 3 groups: (1) acute and chronic hypoxia both low, (2) acute or chronic hypoxia high, or (3) acute and chronic hypoxia both high. The crosstab shows the percent of local recurrences per subgroup in 91 chemoradiotherapy patients. Samples were divided into two groups using the median. Cells are colored in a color corresponding with the line color in the Kaplan–Meier curve. P-value represent the log-rank *p*-value. (**D**) Crosstab showing the percentage local recurrences per subgroup.

**Figure 4 cancers-11-00583-f004:**
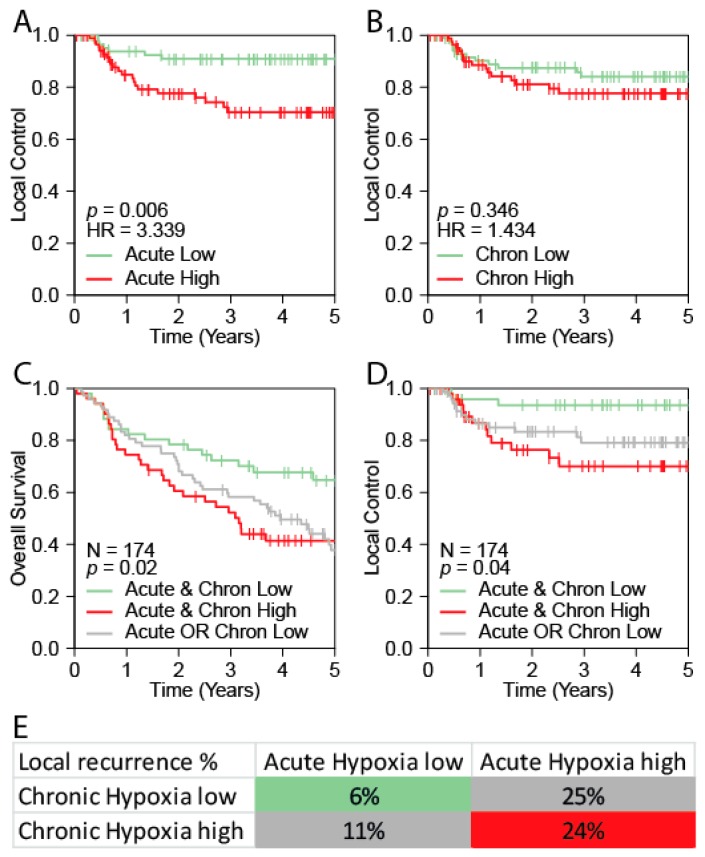
Validation of prognostic value of combined acute and chronic hypoxia scores in the van der Heijden cohort. (**A**) Kaplan–Meier curve showing local control of the “High” versus “Low” acute hypoxia. (**B**) Kaplan–Meier curve showing local control of the “High” versus “Low” chronic hypoxia. (**C**) Kaplan–Meier curve overall survival (**C**) and local control (**D**) for 3 groups: acute and chronic hypoxia both low, acute or chronic hypoxia high or acute and chronic hypoxia both high. (**E**) Crosstab showing the percentage of local recurrences per subgroup in the validation cohort consisting of 174 patients treated with chemoradiotherapy. Cells are colored in a color corresponding with the line color in the Kaplan–Meier curves in panel **C** and **D**.

**Table 1 cancers-11-00583-t001:** Summary of characteristics of the five HNSCC patient cohorts.

Use	Study	Cohorts	*N*	Sites	Treatment	Material	Assay
Combined for classification analyses	Pramana [32]	Stage III-IV HNSCCs	91	All head and neck	Chemo- radiotherapy	FF	Dual channel Operon microarray
	test cohort						
	De Jong 1 [33]	Larynx / oropharynx	99	Larynx/oropharynx	Radiotherapy	FF	Illumina beads microarray
	De Jong 2 [34]	T2-3 larynx	34	Larynx	Radiotherapy	FFPE	RNAseq
Combined for validation cohort	Van der Heijden - NKI-CRAD [35,36]	Stage III-IV HNSCCs	98	Larynx, hypopharynx, HPV-neg oropharynx	Chemo- radiotherapy	FF	RNAseq
	Van der Heijden – DESIGN [35,36]	Stage III-IV HNSCCs	76	Larynx, hypopharynx, HPV-neg oropharynx	Chemo- radiotherapy	FF	RNAseq

FF: Fresh-Frozen; FFPE: Formalin Fixed Paraffin Embedded.

**Table 2 cancers-11-00583-t002:** Multivariable Cox proportional hazard analysis of parameters with patient outcome in the Van der Heijden cohort.

Variable	OS		PFS		LC		LRC		DM	
	HR	*p*-Value	HR	*p*-Value	HR	*p*-Value	HR	*p*-Value	HR	*p*-Value
**Gender**										
Female	0.83	0.04	0.56	0.04	0.16	0.013	0.19	0.004	0.28	0.04
Male	REF		REF		REF		REF		REF	
**Tumor site**										
Oropharynx	REF		REF		REF		REF		REF	
Hypopharynx	0.64	0.09	0.69	0.15	0.96	0.12	0.97	0.94	0.52	0.16
Larynx	0.58	0.9	0.79	0.45	1.27	0.56	1.08	0.86	1.08	0.87
**Disease stage**										
II-III	0.83	0.57	0.78	0.45	3.74	0.037	1.68	0.3	0.38	0.2
IVA-IVB	REF		REF		REF		REF		REF	
**Cumulative cisplatin < 200**										
No	REF		REF		REF		REF		REF	
Yes	2.03	0.003	1.93	0.004	6.4	0.0003	2.7	0.012	0.81	0.64
**Tumor Volume**	1.01	0.000044	1.008	0.0006	1.014	0.034	1.007	0.16	1.006	0.19
**Acute Hypoxia**	4.14	0.021	4.02	0.017	46.4	0.014	30	0.0022	5.5	0.13
**Chronic Hypoxia**	0.83	0.65	1	0.99	0.51	0.48	1.23	0.75	1.26	0.73
**Acute Hypoxia * Tumor Volume**	0.98	0.007	0.98	0.03	0.96	0.12	0.96	0.017	0.98	0.26
**Chronic Hypoxia * Tumor Volume**	1	0.99	0.99	0.7	1	0.98	0.99	0.52	0.99	0.57

OS: Overall Survival; PFS: Progression Free Survival; LC: Local Control; LRC: Locoregional Control; DM: Distant Metastasis Free Survival; HR: Hazard Ratio. *: Interaction between the two variables.

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
