# Peer review of "Acute Hypoxia Profile is a Stronger Prognostic Factor than Chronic Hypoxia in Advanced Stage Head and Neck Cancer Patients"

_cancers, 2019, doi:10.3390/cancers11040583_

Round 1

Reviewer 1 Report

This study examines the effects of hypoxia on the behaviour of head and neck cancers. In particular it focusses on differences in the significance of acute and chronic hypoxia in relation to tumour outcome and prognosis. Published  gene expression profiles are used to associate tumours with acute or chronic hypoxia in vivo and in vitro, and the tumour state was determined for a large cohort of tumours and classified into 4 chronic and 1 acute hypoxia groups. Multivariable statistical analyses indicated that acute hypoxia profiles, alone or combined with chronic profiles, could more consistently predict local recurrence and local control rates. The acute hypoxia profile was found to be significantly associated with local control and patients with acute hypoxia “High” had higher local recurrence rates. A second group of tumours was used to confirm the acute hypoxia profile as a prognostic marker. 

The work presented leads to interesting and apparently valid conclusions concerning the particular importance of acute rather than chronic hypoxia to tumour behaviour. The study is well planned and the conclusions of the work are comprehensively discussed. However, for the average oncologist, the report would benefit greatly from a more considerate and careful presentation of the methods and results  - the text of these sections is very difficult to follow and seems incomplete in places - it often seems necessary to guess what exactly the authors mean?

Author Response

The work presented leads to interesting and apparently valid conclusions concerning the particular importance of acute rather than chronic hypoxia to tumour behaviour. The study is well planned and the conclusions of the work are comprehensively discussed. However, for the average oncologist, the report would benefit greatly from a more considerate and careful presentation of the methods and results  - the text of these sections is very difficult to follow and seems incomplete in places - it often seems necessary to guess what exactly the authors mean?

To improve the readability and to clarify methodolgy we adjusted parts of the results and methods section: 

- The four clinical hypoxia profiles are now introduced in the introduction and are shortly discussed in the results before using these profiles

- The used cohorts are now, prior to using them, briefly introduced in the results section and more extensively in the methods section. This to make clear which datasets are used for the particular analyses.

- We aimed to improve and clarify the origin of the different subsets in the methods section.

Overall we provided carefull proofreading to make sure everything is clear to the reader. We hope that we have adressed your comments satisfactory.

Reviewer 2 Report

One of the common features of head and neck cancer is tumor hypoxia which has an impact on the efficacy of radiotherapy. Hypoxia is expected to have a negative prognostic factor. However, the impact of acute or chronic hypoxia on radiotherapy is not known. In this regard, the authors have described a way to define acute and chronic hypoxia. This field is very controversial and there are many hypothesis proposed by different researchers to classify hypoxia. Authors of this manuscript have accurately mentioned prior literature on this. Authors have shown a way to define acute or chronic hypoxia based on gene expression data. Authors have analyzed 224 patients data with four clinical hypoxia profiles based on gene expression. Authors showed that the four clinical hypoxia profiles are related to chronic hypoxia and not acute hypoxia. Author further claim that acute hypoxia is associated with local recurrence after chemoradiotherapy. There is also mention that acute hypoxia shows a stronger association with patient outcome. Although the idea is not very novel, but authors have used substantially improved method to define chronic vs acute hypoxia. I have some concerns about the study which should be addressed.

Major.

1)      Throughout the manuscript, authors have shown data from literature and the current study. It is very hard to differentiate these two. Authors should re-write the manuscript in a way which clearly separates literature from current work.

2)      The method section in the main manuscript have sections from literature while the supplementary methods are all which is done by authors, so it is not looking appropriate. Again, the author should clearly separate literature from current work.

3)      HPV infection is also an important aspect of head and neck cancer. Although this study is focused on Hypoxia but a brief mention of the impact of HPV in the introduction will help readers to get an alternative view

Minor

1)      Short form for hour and minutes can be used as h or min in method section.

Author Response

I have some concerns about the study which should be addressed.

Major.

1)      Throughout the manuscript, authors have shown data from literature and the current study. It is very hard to differentiate these two. Authors should re-write the manuscript in a way which clearly separates literature from current work.

To adress this comment we adapted the introduction, results and methods section substantially. Since the hypoxia profiles used in this manuscript are derived from literature we already introduce them in the introduction section (rows 81-95). We introduce them in the results section prior to using them (rows 111-112) and we discuss the selection of the profiles in the methods section. We hereby hope to clarify that these are already published profiles and not newly generated.

Second adjustment to adress this comment is the introduction of the used datasets. Before showing the results of the analyses we introduce the different datasets briefly in the results (rows 133-137), together with a table to clarify the different datasets. We aim to make clear that most datasets have been published before, except for the De Jong 2 study.

We also adjusted the methods section by adding more detailed description of the methods for the De Jong 2 cohort(rows 342-352), which is the cohort we sequenced for this study. We refer to the supplementary methods for more extensive methods.

2)      The method section in the main manuscript have sections from literature while the supplementary methods are all which is done by authors, so it is not looking appropriate. Again, the author should clearly separate literature from current work.

As previously described we moved the part where we introduce the gene expression profiles, which are from literature, to the introduction (rows 81-95) and removed it from the methods section.

We clarified which datasets are previously published and which one is new. There is only 1 dataset, De Jong 2, and the supplementary methods are briefly discussed in the methods section now (rows 342-352). We also refer to the suppelemetary methods here for more detailed information.

3)      HPV infection is also an important aspect of head and neck cancer. Although this study is focused on Hypoxia but a brief mention of the impact of HPV in the introduction will help readers to get an alternative view

We added a sentence in the introduction (row 49-53), where we discuss the prognostic relevance of HPV in this tumor population. We also tried to emphasis in row 184-187 that the Van der Heijden cohort only contains HPV-negative HNSCCs to further test whether acute hypoxia is independent of HPV.

Minor

1)      Short form for hour and minutes can be used as h or min in method section.

Adjusted accordingly

Reviewer 3 Report

I think the text needs some minor English changes
I'd place the paragraph of the methods before the discussion to favor understanding
Clinical applicability is very difficult

Author Response

I think the text needs some minor English changes

Adjusted accordingly.

I'd place the paragraph of the methods before the discussion to favor understanding

To our understanding this current format is the preferred format for Cancers. Therefor, we have adjusted the results section, to make it more understandable without having to forward to the methods section.

Clinical applicability is very difficult.

We agree that this method is currently not clinically applicable. Further development is needed to validate and simplify the methods. The message of this study is that acute hypoxia is more important than chronic hypoxia in our reasonably sized cohorts. A large prospective study (NIMRAD) is already validating a (chronic) hypoxia signature. We feel future research (both predictive and prognostic) might be improved with the addition of acute hypoxia.